# Learning Dense Reward with Temporal Variant Self-Supervision

Yuning Wu[1,2], Jieliang Luo[2], Hui Li[2]

*Abstract*— **Rewards play an essential role in reinforcement learning. In contrast to rule-based game environments with well-defined reward functions, complex real-world robotic applications, such as contact-rich manipulation, lack explicit and informative descriptions that can directly be used as a reward. Previous effort has shown that it is possible to algorithmically extract dense rewards directly from multimodal observations. In this paper, we aim to extend this effort by proposing a more efficient and robust way of sampling and learning. In particular, our sampling approach utilizes temporal variance to simulate the fluctuating state and action distribution of a manipulation task. We then proposed a network architecture for self-supervised learning to better incorporate temporal information in latent representations. We tested our approach in two experimental setups, namely joint-assembly and door-opening. Preliminary results show that our approach is effective and efficient in learning dense rewards, and the learned rewards lead to faster convergence than baselines.**

## I. Introduction

Reinforcement learning (RL) is gaining momentum in solving complex real-world robotics problems. One challenging category is contact-rich manipulation tasks. The success of RL in these scenarios depends on a reliable reward system. While this genre of problems is marked by rich, high-dimensional, continuous observations, it is typically hard to come up with a dense reward that can harness such richness to guide RL training. The conventional way of using sparse, boolean rewards (e.g., 1 if the task is successfully completed and 0 otherwise) is often challenging and inefficient. The difficulty has led to the practice of reward engineering, where rewards are hand-crafted based on domain knowledge and trial-and-error. However, such solutions often require extensive robotics expertise and can be quite task-specific.

In this research, we propose an end-to-end learning framework that can extract dense rewards from multimodal observations, inspired by [1]. The reward is learned in a self-supervised manner by combining one or two human demonstrations with a physics simulator, and can then be directly used in training RL algorithms. We evaluate our framework in two contact-rich manipulation tasks, joint-assembly and door-opening tasks.

There are two main contributions in this paper: 1) a temporal variant forward sampling (`TVFS`) method that is more robust and cost-efficient in generating samples from human demonstrations for contact-rich manipulation tasks, 2) a self-supervised latent representation learning architecture that can utilize sample pairs from `TVFS`.

[1]Carnegie Mellon University, Pittsburgh. [2]Autodesk Research, San Francisco. This research was conducted during Yuning Wu's internship at the Autodesk AI Lab and Autodesk Robotics Lab.

## II. Problem Statement & Related Work

### A. Problem Statement

We focus on contact-rich tasks that can be suitably framed as discrete-time Markov Decision Processes (MDPs) [2], which is described by a set of states $S$, a set of actions $A$, a set of conditional probabilities $p(s_{t+1}|s_t, a_t)$ for the state transition $s_t \rightarrow s_{t+1}$, a reward function $R : S \times A \rightarrow \mathbb{R}$, and a discount factor $\gamma \in [0, 1]$. The MDPs can be solved by using RL algorithms to train an optimal policy $\pi(s) \rightarrow a$ that maximizes the expected total reward. Our goal is to learn a dense reward function $R$, which can be used by RL algorithms to reach the optimal policy for the MDPs.

### B. Inverse Reinforcement Learning

To tackle the reward engineering problem, Inverse Reinforcement Learning (IRL) has arisen as a prominent solution [3]. Rather than crafting the reward, IRL methods learn reward functions [4], [5], [6] from expert demonstrations. However, conventional IRL methods mostly deal with ideal scenarios where states and representations are discrete and low-dimensional. Recent advances in deep learning have extended classical IRL's capability to continuous, high-dimensional observation space [7], [8]. However, despite much improved performance, learning is often conducted using generative adversarial learning frameworks [9], [10], meaning that one must train the reward function alongside a policy. Besides instability in training, this framework essentially diverges from our goal, i.e. to learn a reward function independently without concurrently learning a policy.

### C. Learning Dense Reward for Contact-Rich Manipulation

[11] and [12] explored the idea of training a dense reward function directly from human feedback. The methods integrated human experts in the RL training process and periodically ask their preference on a group of pairwise videos clips of the agent's behavior. A reward function is gradually trained that eventually can best explain the human's judgments. The methods showed impressive results on training complex robotic locomotion tasks, but haven't been tested on contact-rich manipulation tasks where the behaviors are hard to be observed by pure visual cues. [1] proposed a `DREM` framework that extracts dense reward from multimodal observation through sampling and self-supervised learning. The framework shows great potential in translating rich, continuous, high-dimensional observations into a task progress variable that can be used to guide RL training. Our work builds on top of this research, by proposing improvements to the sampling method and self-supervised learning architecture. [13] also proposed to learn

a multimodal representation of sensor inputs and use the representation for policy learning on contact-rich tasks, such as peg insertion. However, it uses crafted reward functions for various sub-tasks.

## III. APPROACH: LEARNING DENSE REWARD WITH TEMPORAL VARIANT SELF-SUPERVISION

Similar to ideas presented in [1], our method also aims to learn a task progress variable $p \in [0, 1]$ that captures the progress towards finishing a task. The variable can then be used as a dense reward. With $p = 0$ representing the initial state, and $p = 1$ representing the goal state, the variable is structured as a similarity score in the latent space $\mathcal{H}$. The latent representation $h_\phi : \mathcal{S} \to \mathcal{H}$ is learned in a self-supervised manner with two major objectives. The first is to capture an efficient, low-dimensional embedding of the multimodal observation space $\mathcal{S}$. The second is to encode temporal information in the learned representation. Adopted from [1], for contact-rich manipulation tasks with relatively determined and repeatable goal state, the task progress can be derived with distance measure $d$ in $\mathcal{H}$:

$$p = 1 - \frac{d(h_\phi(s), h_\phi(s_g))}{d(h_\phi(s_0), h_\phi(s_g))} \quad (1)$$

Prior work has explored ways to learn the representation through explicitly enforcing temporal order through a triplet loss function [1]. Such enforcement by design involves tuning multiple hyperparameters. We propose a framework where temporal information can be injected in a more natural, self-consistent manner by utilizing dynamic relation among pairs of adjacent observations $(s_t, s_{t+1})$.

$$h_\phi(s_t) + \Delta h_\psi(s_t) = h_\phi(s_{t+1}) \quad (2)$$

$h_\phi$ is the latent representation, whereas $\Delta h_\psi$ is the change of latent representation between $t$ and $t+1$ resulted from dynamics. $h_\phi$ and $\Delta h_\psi$ are learned using different modalities. The insight behind such constraint is that latent representation of $s_{t+1}$ should be consistent with the representation of $s_t$, plus any dynamic change happened within the time step.

Our proposed improvements are the following. The first is temporal variant forward sampling, which generates a tree of data (i.e. observations with temporal information) from a single human demonstration. The second is self-supervised representation learning network architecture, which uses generated observation pairs to learn representations through Eq.(2).

### A. Temporal Variant Forward Sampling

Collecting observations with temporal information is an essential step for training, but can be challenging if only from human demonstrations. Therefore, the idea of sampling has been broadly experimented in [1], [14]. By combining one or two human demonstrations with a physics simulator, it is possible to obtain a tree of data through sampling. [1] proposed a backward sampling process based on the insight that variance of the goal state is smaller than the initial

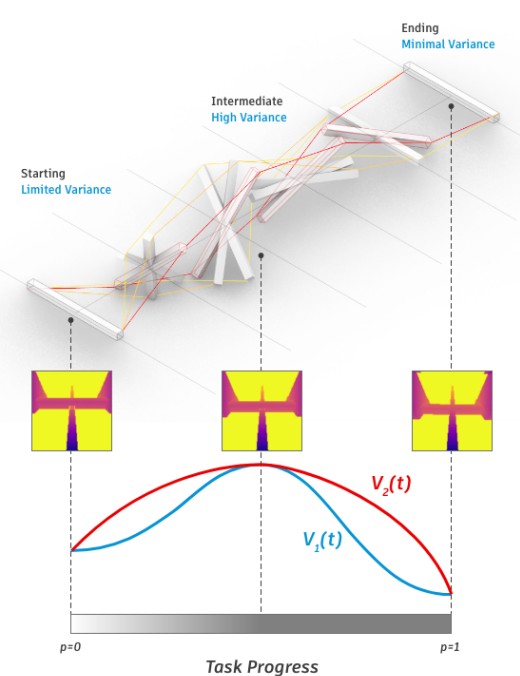

Fig. 1. Sampling variance along different stages of a manipulation task. The blue $V_1(t)$ and red $V_2(t)$ curves show potential temporal variance control functions that can be used in sampling.

state. However, although it is feasible to sample backward positions and generate visual images from the positions in simulation, it is typically hard to sample backward force and torque (F/T). Through experiments, we have found that restoring a state with the exact F/T reading can be computationally intensive and simulator-dependent. Also, when playing *forward* a backward sampled action sequence, the F/T readings in the forward pass do not necessarily match the F/T readings recorded in the backward pass.

Without loss of generality, we propose a new sampling process, named Temporal Variant Forward Sampling (TVFS) that aims to tackle the aforementioned challenges, while capturing the fluctuating variance of manipulation tasks. The insight behind our method is to roughly control sampled actions with a temporal variance $V(t)$, such that sampled actions do not diverge too much from the potential distribution of an expert demonstration, and that the actions are mostly progressing forward. For instance, an action that is the opposite of an expert action may not appear at certain stable stages. As shown in Fig.1, at the starting stage ($p = 0$), the sampling variance is limited. At the intermediate stage, the variance is high due to lack of constraints and high moving flexibility. At the ending stage ($p = 1$), the variance is low because the goal state is relatively deterministic. $V(t)$ can be depicted with a chosen kernel function. In our experiment, we have chosen the quadratic function for simplicity. The general process of our sampling method is illustrated in Fig.2, and is described as follows.

1) We record an expert demonstration in simulation, and

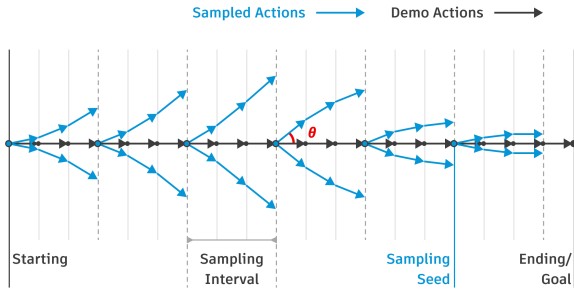

Fig. 2. Temporal variant forward sampling (TVFS). The difference between sampled actions (blue) and demo actions (black) are controlled through $V(t)$, which is also a variance measure that changes along the task process.

choose the sampling seeds (states) $\{Q_0, Q_1, \cdots, Q_M\}$ by certain sampling interval.

2) At each seed (state) $Q_i$, randomly sample $N$ branches. Each branch may contain multiple forward steps. At each step, control the variance between sampled action $a_t^{\text{sampled}}$ and demo action $a_t^{\text{demo}}$. The variance can be measured using any similarity score. In our case, we have chosen the cosine similarity.

3) Record the sampled observations in pairs $(s_t, s_{t+1})$, such that they can be used in learning Eq.(2).

### B. Multimodal Representation Learning

With the generated multimodal observation pairs from TVFS, we have designed a network architecture and a loss function to incorporate temporal information in representation learning. As mentioned above, we use different modalities to learn different components of Eq.(2). $h_\phi$ is learned with static modalities such as images, depth maps and poses, while $\Delta h_\psi$ is learned using dynamic modalities such as F/T and velocities. The two separately learned components should be consistent in the latent space. We accentuate such consistency with a hybrid loss function, consisting of *temporal enforcement loss* and reconstruction loss. The architecture and loss functions are detailed as follows.

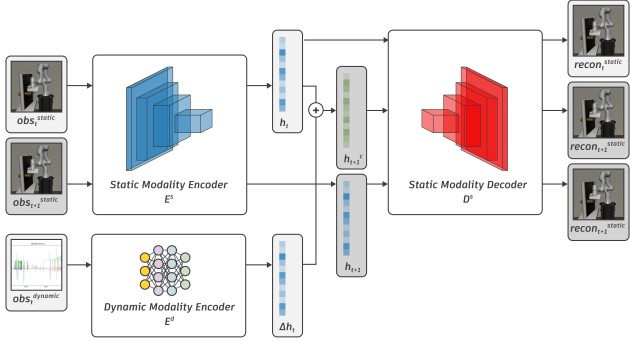

Fig. 3. Self-supervised learning network architecture

1) *Static Modality Encoder* $\mathbf{E}^{\mathbf{s}}$. To learn $h_\phi$, we use a fixed RGB-D camera as input for static modality

encoders. The RGB image (256x256x3) and depth map (256x256x1) are handled separately. Similar to [13], the network is composed of a 6-layer Convolutionary Neural Network and a fsully-connected layer. Depending on the experiment scenarios, one may switch or combine the modalities. An extra Multi-Layer Perceptron may for output fusion. The final embedding is a 64 dimensional hidden vector.

2) *Dynamic Modality Encoder* $\mathbf{E}^{\mathbf{d}}$. To learn $\Delta h_\psi$, we use F/T reading and velocity as input for the dynamic modality encoder. Due to the accumulative nature of F/T, we use a window size of 32 to better capture the momentum. The 32x6 input is passed into a 4-layer Causal Convolution Network. The output is concatenated and fused with velocity to produce another 64 dimensional hidden vector.

3) *Static Modality Decoder* $\mathbf{D}^{\mathbf{s}}$. Through experiments, we have found that instead of enforcing self-supervised learning on both static and dynamic modalities, it is better to focus on one side only. This choice will be explained further in later descriptions. In our case, we are proposing an auto-encoding architecture on the static modality side. The decoder takes input of a 64 dimensional hidden vector and use transposed Convolutional Neural Network to reconstruct the RGB image / depth map.

4) *Latent Representation Learning*. As mentioned briefly in previous context, the temporal order is injected through learning with pairs of adjacent observations $(s_t, s_{t+1})$. By enforcing Eq.(2) among each pair, the temporal relation among latent representations are broadcasted. Fig.3 is an illustration of the whole network architecture. We first encode and decode $s_t$ with $\mathbf{E}^{\mathbf{s}}$, $\mathbf{E}^{\mathbf{d}}$ and $\mathbf{D}^{\mathbf{s}}$, then use the embedding to craft an latent representation for $s_{t+1}$.

$$\hat{h}(s_{t+1}) = \mathbf{E}^{\mathbf{s}}(s_t) + \bar{\mathbf{E}}^{\mathbf{d}}(s_t) \tag{3}$$

The hybrid loss function is structured around $\hat{h}(s_{t+1})$. The first component is *temporal enforcement loss*, which enforces Eq.(2) in the latent space. To ensure effectiveness of $\Delta h_\psi$, we are applying $L2$ normalization to $\bar{\mathbf{E}}^{\mathbf{d}}(s_t)$.

$$l^{\text{temporal}} = \text{MSE}\left[\hat{h}(s_{t+1}), \mathbf{E}^{\mathbf{s}}(s_{t+1})\right] \tag{4}$$

The second component is reconstruction loss, which provides supervision signal to the auto-encoding architecture. As apposed to directly decoding $\mathbf{E}^{\mathbf{s}}(s_{t+1})$, we are decoding $\hat{h}(s_{t+1})$ so that Eq.(2) is also enforced in self-supervision.

$$l^{\text{recon}} = l^{\text{recon}}(s_t) + \hat{l}^{\text{recon}}(s_{t+1}) \tag{5}$$

$$l^{\text{recon}}(s_t) = \text{MSE}\left[\mathbf{D}^{\mathbf{s}}(\mathbf{E}^{\mathbf{s}}(s_t)), s_t\right] \tag{6}$$

$$\hat{l}^{\text{recon}}(s_{t+1}) = \text{MSE}\left[\mathbf{D}^{\mathbf{s}}\left(\hat{h}(s_{t+1})\right), s_{t+1}\right] \tag{7}$$

The two loss components are then combined through

a hyperparameter $\lambda$.

$$l = l^{\text{recon}} + \lambda \cdot l^{\text{temporal}} \tag{8}$$

We set $\lambda = 10$ in training to accentuate the temporal relation, so that the representation learning does not converge suboptimally too early. The learned embedding is then used in Eq.(1) for the dense reward.

## IV. Experimental Results

The experiments are conducted in simulation. In order to test that our sampling method can be generalized to different simulators and robot controllers, we tested lap-joint assembly in PyBullet [15] with an robot-agnostic environment, and door-opening in Robosuite [16] with a Panda robot. We conducted similar sampling process on both tasks, setting sampling interval $I = 50$, number of branches $N = 5$, number of steps per branch $K = 10$. The temporal variance is controlled in $\theta \in \left[\frac{\pi}{12}, \frac{\pi}{4}\right]$. We trained the model with an NVIDIA 3060 GPU for around 5,000 iterations. Compared to the training iterations mentioned in [1], our method is potentially more efficient. We defer ablation study and further examination of this comparison to future work.

### A. Visualization of Learned Dense Reward

We visualized the learned dense reward in two cases. The results indicate that the dense reward learned by our approach is effective. In the first case (Fig.4), we compare the rewards between a successful trajectory and a failed trajectory in the lap-joint task. The plots suggest that a successful trajectory has rewards gradually increasing from 0 to 1, which matches the definition of task progress. A failed trajectory have a decreasing reward dropping below 0, which can happen when the agent get into unexpected scenarios.

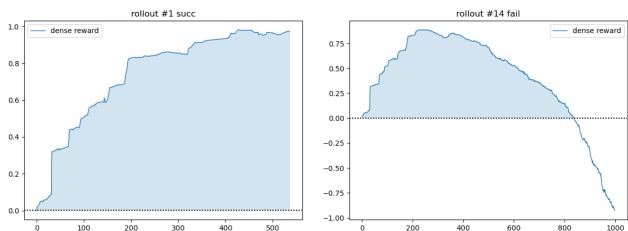

Fig. 4. Comparison of a successful trajectory (left) and a failed trajectory (right). Visualization produced in the lap-joint task.

In the second case (Fig.5), we examine rewards of an inexpert demonstration in door-opening. The demonstrator experienced a plateau of trial-and-error when rotating back and forth the door handle. While the hand-crafted reward mostly gives analogous signals during this period, our rewards provides fluctuations indicating more detailed feedback for learning.

### B. Dense Rewards for Policy Training

To examine the performance of the learned reward in policy training, we have chosen Soft Actor-Critic [17] as the RL algorithm for benchmarking. We compared three types of

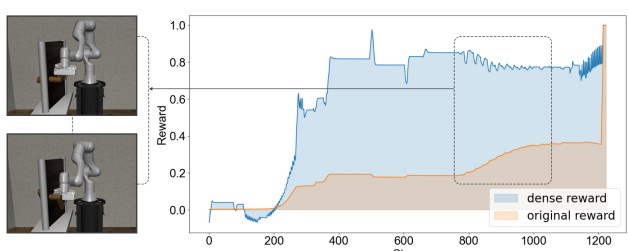

Fig. 5. Comparison of dense reward and hand-crafted rewards in door opening task

rewards in the door-opening task, namely our dense reward, a hand-crafted reward based on distance ($\gamma \|x_t - x_g\|_2$), and the sparse boolean reward. In particular, we trained the policy for door-opening task for 500 epochs. For each type of reward, we conducted three training experiments with different random seeds. The results (Fig.6) indicate that our dense reward leads to faster convergence, and more training stability.

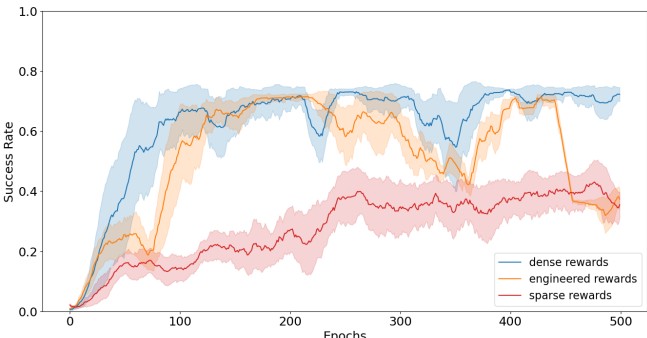

Fig. 6. RL training comparison among three types of rewards: our dense reward (blue), hand-crafted distance reward (orange), and sparse boolean reward (red).

## V. Conclusions and Future Works

In this paper, we propose an improved framework for learning dense reward for contact-rich manipulation tasks. The framework includes a more robust sampling method, namely temporal variant forward sampling (TVFS), that can generate samples from one or two human demonstrations with a physics simulator. A self-supervised learning architecture is also designed to efficiently utilize the generated sample pairs.

For future work, we intend to conduct more ablation studies regarding the framework's adaptability and modalities. For instance, during experiments we observe that camera setup can have a substantial impact on the learning result. Therefore one potential is to study whether we can mainly rely on pure tactile sensors for reward inference. Another potential is to test whether the reward can be transferred to manipulation tasks with nondeterministic goal state.

ACKNOWLEDGEMENT

We thank Tonya Custis and Sachin Chitta for budgetary support of the project; Yotto Koga for simulation support; our colleagues at Autodesk Research for the valuable feedback, and Zheng Wu for the discussions.

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
