# OpenReview forum: "Learning Dense Reward with Temporal Variant Self-Supervision"
_ICRA.org/2022/Workshop/Contact-Rich — ICRA 2022 Workshop: RL for Manipulation Poster_

### Official Review · Reviewer_eaTz · 2022-05-04
**Review of "Learning Dense Reward with Temporal Variant Self-Supervision"**

**Rating:** 7
**Confidence:** 4

**Review:**

Overall:
In this paper the authors proposed an extended version of [1] for learning dense rewards from multimodal high-dimensional sensor input. The main contributions of this work are a different data sampling method and a different self-supervised learning architecture. The experimental results show the proposed learned reward outperforms sparse reward and engineered reward in two contact-rich manipulation tasks.

Strengths:
The paper is clearly written and easy to understand.
The proposed method is technically sound. The sampling method can potentially tackle the problem of F/T reading during reversed sampling proposed in [1].
The visualization results indicate the rationalness of the learned reward in both tasks and benefit RL training compared to engineered reward and sparse reward.
Weaknesses:
More experiments need be conducted to validate the effectiveness of the proposed method. The authors proposed some improvements over [1] but the experiment section lacks the comparison with [1].
Caption of Figure 3. in the paper should be more detailed and the reconstructed images in Figure 3. seems to be the original images instead of reconstructed output.



[1] Z. Wu, W. Lian, V. Unhelkar, M. Tomizuka, and S. Schaal, “Learning dense rewards for contact-rich manipulation tasks,” 2021 IEEE International Conference on Robotics and Automation (ICRA), pp. 6214–6221, 2021.

---

### Official Review · Reviewer_WiPY · 2022-05-08
**Interesting addition for learning dense rewards by leveraging structured latent spaces with multi-modal inputs**

**Rating:** 8
**Confidence:** 5

**Review:**

This paper proposes a self-supervised learning approach to improve the efficiency and robustness of learning dense rewards for reinforcement learning (RL) tasks.
The core idea is to introduce temporal variances from a small set of human demonstrations to encode a task progress variable.
This variable provides information on the task progress (a value of 0 corresponds to initial state and 1 of the goal state) that is computed using a latent space embedding of the task's multimodal observations (e.g., vision and force/torque data).
The authors conduct several experiments and compare their approach with baselines.
Preliminary results indicate that the proposed approach improves learning of RL tasks in various benchmarks.
The presented idea of improving the latent space embedding is interesting and builds upon the foundations of a recently proposed paper that introduced the task progress variable.
The paper is well written and easy to understand, since the authors nicely motivate each choice in their algorithm.
However, the paper can be further improved in several areas to better highlight the approach.
The paper relies on a set of human demonstrations of the task.
Yet, the paper does not really outline how these human demonstrations are obtained.
For instance, are they recorded on a real system using demonstrations or in simulation?
What happens if the humans provide uncertain feedback?
Moreover, how does the approach deal with demonstrations that end in different goal states?
How many demonstrations are needed and with what kind of quality (e.g., regarding the different modalities)?
Secondly, it would be beneficial if the reader can get more details on the sampling approach.
So far, the paper mostly outlines the high-level idea of temporal variance but does not really provide details to reproduce the idea.
For instance, how is the tree generated exactly?
Are the samples uniformly created?
How is the variance computed and integrated into the sampling?
How does the sampling interact with human demonstrations that vary from sample to sample?
Lastly, the paper would strongly benefit from a limitation section, describing current caveats of the approach.
The paper describes that there is usually one distinct goal state for the task that can be used to compute the task progress.
However, what if this is not the case and there are multiple goal states?
Can one also incorporate goal regions?
How is the distance compute for complex contact-rich manipulations tasks where the robot might need to reposition the end-effector, resulting in increasing the distance to the goal again?
Lastly, how is noise incorporated into the approach?
Especially, force/torque readings are noisy.
Regarding the style of the paper, Figure 1 is hard to read, especially the variance plots.
Can the authors find a better way to illustrate the sampling?
This figure is the essential one in the paper.
The bibtex needs a revision, e.g., [10] and [11] miss pages and [3] has incorrect pages and needs to capitalize ICML.
In summary, this paper presents a very interesting idea that is highly relevant to the goals of the workshop.